# The Correlation between Chinese Written Vocabulary Size and Cognitive, Emotional and Behavioral Factors in Primary School Students

**DOI:** 10.3390/ijerph18157797

**Published:** 2021-07-22

**Authors:** Ning Pan, Yangfeng Guo, Jingwen Ma, Xiaoxuan Fan, Zhixin Yin, Xiaoyu Xu, Lei Cai, Yue Zhang, Xiuhong Li

**Affiliations:** 1Department of Maternal and Child Health, School of Public Health, Sun Yat-Sen University, Guangzhou 510080, China; pann3@mail2.sysu.edu.cn (N.P.); 13544333684@163.com (J.M.); fanxx5@mail2.sysu.edu.cn (X.F.); a982004766@126.com (Z.Y.); xuxy39@mail2.sysu.edu.cn (X.X.); chuxinbest@live.com (L.C.); 2Department of Prevention and Treatment for Common Disease, Guangzhou Health Promotion Center for Primary and Secondary School, Guangzhou 510080, China; guoyangfeng2021@126.com; 3Children’s Health Care Department, National Center for Women and Children’s Health, Chinese Center for Disease Control and Prevention, Beijing 100081, China; zhangyue0416@163.com

**Keywords:** Chinese character, written vocabulary size, emotional factor, behavioral factor, child

## Abstract

Written vocabulary size plays a key role in children’s reading development. We aim to study the relationship between Chinese written vocabulary size and cognitive, emotional, and behavioral factors in primary school students. Using stratified cluster sampling, 1162 pupils from Grade 2~5 in Guangzhou were investigated. Chinese written vocabulary size, cognitive, emotional, and behavioral factors were assessed by the Chinese written vocabulary size assessment scale, the dyslexia checklist for Chinese children (DCCC) and the Strengths and Difficulties Questionnaire (SDQ), respectively. The scores of visual word recognition deficit (*β* = −3.32, 95% CI: −5.98, −0.66) and meaning comprehension deficit (*β* = −6.52, 95% CI: −9.39, −3.64) were negatively associated with Chinese written vocabulary size; the score of visual word recognition deficit (odds ratio (OR) = 1.04, 95% CI: 1.02, 1.07) was the related factor of a delay in written vocabulary size. The score of meaning comprehension deficit was negatively associated with boys’ Chinese written vocabulary size, while the score of auditory word recognition deficit was negatively associated with girls’ Chinese written vocabulary size. The related factor of a delay in written vocabulary size was spelling deficit in boys and visual word recognition deficit in girls. There is a significant correlation between Chinese written vocabulary size and cognitive factors, but not emotional and behavioral factors in primary school students and these correlations are different when considering gender.

## 1. Introduction

Written vocabulary knowledge includes the number of words known (breadth or vocabulary size) and richness of knowledge about the words known (depth) [1]. Chinese written vocabulary size refers to the number of Chinese characters known by school-age children in our study. It is the foundation of reading activities and a representative indicator of children’s reading development, which is highly correlated with academic performance [2]. The delay in written vocabulary size prevents children from reading correctly and fluently, which further leads to academic failure and loss of confidence in learning [3]. The primary school stage is a critical period for written vocabulary size acquisition [4]. The delay in written vocabulary size in primary school is a powerful predictor of the difficulties of reading and writing in adolescents and adults [5,6].

Many factors may have an association with written vocabulary size, including family reading environment, school teaching quality, children’s characteristics, and so on. In this study, we gained more insight into the relationship between written vocabulary size and children’s characteristics, including cognitive, emotional, and behavioral factors. Word learning is a complex cognitive processing that requires a lot of cognitive skills to participate in, for example, visual processing, phonological processing, meaning comprehension, spelling, memory and attention, and so on [7,8]. So far, it is still controversial which cognitive factors are the core factors that affect word learning. In alphabetic writing systems, phonological processing is considered to be the core ability for word learning [9]. Interestingly, different from the alphabetic writing system, Chinese characters are a kind of ideographic writing system, so Chinese learning may be more dependent on visual processing, semantic processing, and orthographic processing [10]. Different studies have used different cognitive factors; therefore, it is still controversial which is the main cognitive factor affecting Chinese written vocabulary size.

In terms of emotional and behavioral factors, studies showed that there is a consensus about substantial comorbidity between vocabulary delay and emotional and behavioral problems in school-age children [11,12]. Children with emotional disorders, hyperactivity, or conduct problems also suffer from a delay in written vocabulary size of both alphabetic and logographic writing systems [13,14,15]. A possible mechanism may be that negative emotional expressions and hyperactivity may hinder attentional processes and overburden the cognitive resources which are needed for word learning [16]. Literature showed that cognitive and emotional processing interacted with each other and had a joint influence on behavior pattens [17,18,19], so it was reasonable when exploring the influencing factors on written vocabulary size that cognitive, emotional, and behavioral factors should be considered at the same time.

In addition, we also focused on the gender difference in correlation with Chinese written vocabulary size. There are significant differences in cognitive abilities between boys and girls [20]. Boys have an advantage in visuospatial information processing [21], while girls perform better in language comprehension and expression [22]. At the same time, children of different genders also suffer from different emotional and behavioral problems in primary school. Girls tend to show emotional and peer relationship problems, while boys are more likely to represent hyperactivity and conduct problems [23]. However, evidence for the gender difference in correlation with written vocabulary size is much less, especially in the Chinese written system.

To summarize, little empirical evidence exists considering multidimensional factors related to Chinese written vocabulary size. Here, we aimed to explore: 1) which cognitive, emotional, and behavioral factors are the main factors influencing Chinese written vocabulary size after controlling for demographic information. In our study, cognitive factors included visual word recognition, auditory word recognition, meaning comprehension, spelling, oral language, written expression, and attention; emotional factors mainly included anxious and depressive emotional symptoms; behavioral factors included conduct problems, hyperactivity/inattention, peer relationship problems, and prosocial behavior. 2) Whether the influencing factors have different patterns between boys and girls. Solving the two questions can provide a comprehensive reference for improving Chinese written vocabulary size instruction of children in primary schools and children with dyslexia in clinics.

## 2. Materials and Methods

### 2.1. Study Design

This study was conducted from October 2016 to March 2017 in Guangzhou, Guang-dong Province of China. Using the method of stratified cluster sampling, we selected five public primary schools in five districts of Guangzhou and investigated all the students in Grade 2–5 of all the primary schools. A total of 2057 children from Grade 2 to 5 were enrolled. They were all native Chinese speakers and learned English as their second language since entering primary school. Among them, 2026 questionnaires were returned and the rate of recovery was 98.5%. In total, 864 cases were excluded because of missing data for the main assessments. Finally, there were 1162 children entered into the analysis, including 606 boys and 556 girls. The investigators were experienced graduated students who received professional training on the survey. None of the subjects reported intellectual disability, autism spectrum disorder, or attention deficit hyperactivity disorder.

This study was supported by the Key Realm R&D Program of Guangdong Province (grant number 2019B030335001), the Guangdong Basic and Applied Basic Research Foundation (grant number 2021A1515011757), and the National Natural Science Foundation of China (grant number 81673197). This study was approved by the Ethics Committee of the School of Public Health, Sun Yat-sen University (L2016-036). All parents of the children signed informed consent before their inclusion in our study.

### 2.2. Data Measurement

#### 2.2.1. Participant Demographics

A self-designed questionnaire was used to collect the demographic information. The content consists of children’s gender, date of birth, school, grade, only one child or not, mode of delivery, parents’ education level, and family income.

#### 2.2.2. Chinese Written Vocabulary Size Assessment Scale for Primary School Children

The Chinese written vocabulary size assessment scale was compiled by Wang and Tao of East China Normal University [24]. This scale was widely used to assess Chinese written vocabulary size in Grade 2–5 [25]. There were 10 sets of questions in the test paper, and each group had 6 to 33 Chinese characters. Written vocabulary size, which was at least 1.5 standard deviation (SD) below the average level of the actual grade, represented the existence of a delay in written vocabulary size. This standardized test had a reliability and validity of 0.98.

#### 2.2.3. The Dyslexia Checklist for Chinese Children (DCCC)

The DCCC, established by Wu in 2006, was used to assess the cognitive abilities of Chinese students in Grade 2–5 [26]. It contained 57 items and synthesized 8 factors, including the deficit of visual word recognition, the deficit of auditory word recognition, the deficit of meaning comprehension, the deficit of spelling, the deficit of oral language, the deficit of written expression, bad reading habits, and the deficit of attention. In this study, seven factors other than bad reading habits were used to evaluate children’s cognitive abilities. The higher score of each factor indicated more serious difficulties in cognitive abilities. The test–retest reliability of the DCCC was 0.734, and the internal consistency of all subscales was above 0.752 [27].

The meaning of each factor is as follows: (1) The deficit of visual word recognition mainly refers to children’s difficulties in the visual processing of Chinese characters. (2) The deficit of auditory word recognition mainly refers to children’s difficulties in the phonological processing of Chinese characters. (3) The deficit of meaning comprehension mainly refers to children’s difficulties in the acquisition and processing of semantic access in different levels, including characters, vocabularies, sentences, paragraphs, and texts. (4) The deficit of spelling mainly refers to children’s poor fluency and recognizability of writing. (5) The deficit of oral language mainly refers to children’s difficulties in oral comprehension and oral expression. (6) The deficit of written expression mainly refers to children’s difficulties in using and outputting written language, reflecting children’s comprehensive obstacles in meaning processing. (7) Bad reading habits mainly include reading the same line repeatedly, skipping characters, losing characters, adding characters, and making a sound when children are reading. (8) The deficit of attention mainly refers to children’s difficulties in attention and concentration levels.

#### 2.2.4. The Chinese Version of the Strengths and Difficulties Questionnaire Rated by the Parent (SDQ)

The SDQ was designed to identify children’s emotional and behavioral problems [28]. It contained 25 items and was divided into 5 subscales, including emotional symptoms, conduct problems, hyperactivity/inattention, peer relationship problems, and prosocial behavior. The higher scores of the difficulty part were indicated to have more serious emotional and behavioral problems, except for the prosocial behavior score, where a lower score indicated greater difficulties. The retest stability was 0.564~0.772 and the content validity was 0.482~0.774 in Chinese children [29].

### 2.3. Statistical Analysis

Data were entered using EpiData3.0 and statistics were performed using SPSS 23.0. Descriptive statistics were applied to present the characteristics of participants’ demographics and the chi-square test was used to assess the difference between boys and girls. Two-sample *t*-test, chi-square test, one-way ANOVA, and Pearson correlation analysis were used for univariate analysis. Multiple stepwise linear and logistical regression analyses were used to explore related factors of Chinese written vocabulary size and a delay in written vocabulary size, respectively. Participant demographics entered the first regression model (Model 1). Model 2 comprised Model 1, and four difficulties and one strength assessed using SDQ. Finally, Model 3 encompassed Model 2 and the seven factors of DCCC. Then, we performed a hierarchical subgroup analysis of gender. A *p* < 0.05 indicated a statistically significant difference.

## 3. Results

### 3.1. Demographic Distribution of the Subjects in Five Primary Schools

Table 1 shows the demographic information of the students in Grade 2 to 5 of five public primary schools. The average age was 9.19 ± 1.15 years and the sex ratio was 1.09:1 (boys: girls). Most parents’ education levels were a senior middle school or above. The per capita annual income of most pupils’ families was 3000–15,000 RMB. No statistical difference in demographic information was detected between boys and girls.

### 3.2. Univariate Analysis for Related Factors on Chinese Written Vocabulary Size

About 7.3% of the participating children exhibited a delay in written vocabulary size, and the proportion of boys was significantly higher than girls. The scores of conduct problems, hyperactivity/inattention, and the seven factors of DCCC were significantly negatively correlated with Chinese written vocabulary size. Compared to the children with normal written vocabulary size, the children with a delay in written vocabulary size had higher scores in hyperactivity/inattention, peer relationship problems, and seven factors of DCCC. The results are shown in Table 2.

### 3.3. Multiple Linear and Logistic Regression Analysis for Related Factors of Chinese Written Vocabulary Size

In Table 3, Model 1 showed that grade (*β* = 617.22, 95% CI: 599.26, 635.17) and father’s education level (*β* = 48.40, 95% CI: 22.31, 74.49) were positively associated with Chinese written vocabulary size. Model 2 of Chinese written vocabulary size found that the scores of hyperactivity/inattention (*β* = −21.93, 95% CI: −30.56, −13.30) and peer relationship problems (*β* = −17.41, 95% CI: −30.15, −4.66) were negatively related to Chinese written vocabulary size. Model 3 of Chinese written vocabulary size showed that the scores of the deficit of visual word recognition (*β* = −3.32, 95% CI: −5.98, −0.66) and the deficit of meaning comprehension (*β* = −6.52, 95% CI: −9.39, −3.64) had a significant negative association with Chinese written vocabulary size. However, the emotional and behavioral factors were no longer statistically significant in Model 3. Model 1 of a delay in written vocabulary size showed that boys and children with cesarean birth were more likely to have a delay in written vocabulary. In Model 2 of a delay in written vocabulary size, children with more serious hyperactivity/inattention (OR = 1.18, 95% CI: 1.07, 1.30) and peer relationship problems (OR = 1.15, 95% CI: 1.00, 1.32) were more likely to suffer from a delay in written vocabulary size after controlling demographic information. When further considering cognitive abilities, the degree of a deficit on visual word recognition (OR = 1.04, 95% CI: 1.02, 1.07) was the related factor to a delay in written vocabulary size, and the emotional and behavioral factors had no statistical significance in Model 3.

The results of the subgroup analysis by gender are shown in Table 4. The score of meaning comprehension (*β* = −8.96, 95% CI: −12.27, −5.65) was closely related to boys’ Chinese written vocabulary size, and the degree of a deficit on spelling (OR = 1.04, 95% CI:1.01, 1.07) was correlated to boys’ delay in written vocabulary size in Model 3, which further considers the seven factors of DCCC. The score of the deficit of auditory word recognition (*β* = −8.86, 95% CI: −12.44, −5.29) had an association with girls’ Chinese written vocabulary size, and the degree of a deficit on visual word recognition (OR = 1.07, 95% CI: 1.02, 1.12) was correlated to girls’ delay in written vocabulary size in Model 3.

## 4. Discussion

This is the first multivariate study to assess the relationships between Chinese written vocabulary size and cognitive, emotional, and behavioral factors in primary school students. The main findings included: first, when the cognitive, emotional, and behavioral factors were considered at the same time, only the cognitive factors had a significant correlation with Chinese written vocabulary size in primary school students, including visual word recognition and meaning comprehension; second, the factors related to Chinese written vocabulary size had significant differences between boys and girls.

### 4.1. The Correlation between Chinese Written Vocabulary Size and Cognitive, Emotional, and Behavioral Factors

This study indicated that, after controlling demographic characteristics, the scores of the hyperactivity/inattention and peer relationship problem had significant correlations with Chinese written vocabulary size. In western countries, studies showed that hyperactivity/inattention symptoms during early and middle childhood can predict written vocabulary size at age 12 years [16,30]. In China, hyperactivity/inattention symptoms are also found to be negatively associated with written vocabulary size in Grade 2 to 5 [15]. When learning Chinese characters, children require strong behavioral self-regulation and attention control [31]. The children with more serious hyperactivity/inattention are prone to be distracted by irrelevant stimuli, which may inevitably affect the quality of memory and lead to a delay in written vocabulary size [32]. In line with previous literature, Chinese written vocabulary size also has a strong negative relationship with peer relationship problems [15]. Compared with children who have a negative relationship with their peers, children who have a positive peer relationship appear to have a larger written vocabulary size and are less likely to have a delay in written vocabulary size in our study. Good peer relationships can provide a source of companionship and emotional support for school-age children [33], which are conducive to the development of Chinese written vocabulary size. However, when the cognitive abilities were considered at the same time, the significant correlation between the scores of hyperactivity/inattention and peer relationship problem, and Chinese written vocabulary size disappeared. The results indicated that cognitive factors were the main factors affecting Chinese written vocabulary size, while emotional and behavioral problems were not. Literature showed that cognitive and emotional processing interacted with each other and had a joint influence on behavior pattens [17,18,19], so we speculated that the cognitive factors were the most important factors that directly affect Chinese written vocabulary size, while emotional and behavioral factors are indirect influencing factors.

The most important finding in the current study was that visual word recognition and meaning comprehension had a significant association with Chinese written vocabulary size when controlling demographic characteristics, and emotional and behavioral problems. Visual word recognition mainly refers to children’s difficulties in the visual processing of Chinese characters in the scale of DCCC. Previous studies have suggested that auditory word recognition was the core correlate of written vocabulary size in alphabetic language [34,35]. However, different from alphabetic language, Chinese character is a two-dimensional visual processing unit composed of strokes and has a more complex visual–spatial structure [36]. A few studies indicated that independent of phonological and morpheme skills, visual processing had an important influence on Chinese written vocabulary size, which supported our results [37]. Furthermore, we also found that children with a deficit in visual word recognition were more likely to suffer from a delay in Chinese written vocabulary size. Previous studies indicated that children’s visual skill deficit may lead to insufficient processing of Chinese characters [38], and visual skill can be used to distinguish between children with and without developmental dyslexia in Chinese [39].

In the current study, meaning comprehension mainly refers to the acquisition and processing obstacles of children’s semantic access in different levels, including characters, vocabularies, sentences, paragraphs, and texts in the scale of DCCC. We found that children with a more serious meaning comprehension deficit were more likely to have lower Chinese written vocabulary size. Consistent with our results, some studies indicated that insufficient knowledge of word meaning is a crucial barrier to Chinese written vocabulary size growth [40,41]. Meanwhile, previous studies demonstrated that Chinese written vocabulary size also make an important contribution in meaning comprehension performance due to the children needing to recognize a large number of words to read fluently [40,42]. It seems that children’s meaning comprehension had a bidirectional association with Chinese written vocabulary size, which suggests that the rich and direct instruction of meaning comprehension may be an effective approach to improve Chinese written vocabulary size for primary school students.

### 4.2. The Gender Difference Correlates between Chinese Written Vocabulary Size and Cognitive, Emotional, and Behavioral Factors

Another important finding of our study is that there are significant gender difference correlates between Chinese written vocabulary size and cognitive, emotional, and behavioral factors.

For boys, the cognitive factors influencing Chinese written vocabulary size and a delay in written vocabulary size were meaning comprehension and spelling. First, there is a notable gender difference in meaning comprehension [43]. Compared with girls, boys typically rely more on analysis and logical reasoning for cognitive processing, and prefer to guess the meaning of single characters according to the context [44,45]. The ability of meaning comprehension is the basis of recognizing Chinese character form [46]. Boys with a deficit in meaning comprehension were more likely to have a lower Chinese written vocabulary size. Second, the deficit of spelling mainly refers to children’s poor fluency and recognizability of writing, which reflects the deficit of the encoding process of literacy in our study. The previous study also showed that boys always performed worse than girls at all grade levels on spelling tests, which suggested that the degree of a deficit in spelling ability may be more sensitive to boys [47]. In addition, the learning of Chinese characters can be acquired by copying repeatedly [48]. If boys have poor spelling ability, it will seriously affect their Chinese character recognition and lead to a delay in written vocabulary size. Thus, we speculate that the deficit in meaning comprehension and spelling of boys may give an inverse contribution to the cognitive processing of Chinese characters and relate to Chinese written vocabulary size.

Different from boys, for girls, the cognitive related factor influencing Chinese written vocabulary size was auditory word recognition The literature showed that phonological memory was a significant predictor of the reading and writing ability of Chinese characters [49]. Moreover, girls generally scored higher than boys on phonological memory [50]. We speculate that girls may rely more on the phonological processing ability, such as phonological memory, to recognize Chinese characters. Thus, girls are more likely to have lower Chinese written vocabulary size when their auditory word recognition is poor. In addition, we found that the delay in written vocabulary size was significantly correlated with the degree of a deficit in visual word recognition among girls. Chinese character recognition needs not only auditory processing ability, but also visual processing ability [50]. According to our results, it is inferred that the cognition of Chinese characters is more dependent on visual processing ability, and girls who lag in visual processing ability are more likely to suffer from a delay in written vocabulary size, even if their auditory processing ability is normal. Future work should develop targeted methods based on different genders in both school and clinic written vocabulary size instruction.

### 4.3. The Correlation between Chinese Written Vocabulary Size and Parental Education Level

Interestingly, we found that the father’s education level was positively associated with children’s written vocabulary size. Inconsistent with our results, most studies indicated that there was also a significant association between mother’s education level and children’s vocabulary development [51,52]. However, recent studies indicated that children with a higher father’s education level also have a larger written vocabulary size [41,53]. The further stratified analysis showed that the education level of the father only had an association with the written vocabulary size of boys, but had no association with girls. The possible reasons were still unclear. We speculated that, first, this significant gender difference may reflect a Y-linked inheritance pattern of Chinese written vocabulary size, although the specific genetic pathway remains uncertain. Literature also indicated that the Y chromosome has a significant effect on learning performance by affecting multiple cognitive abilities, such as visuospatial abilities, which play an important role in the developing of Chinese written vocabulary [54,55]. Second, previous studies suggested that fathers were the representative of male behavior patterns in their children’s lives and are the main role models for boys’ role identification [56,57,58]. Boys were more likely to portray fathers as their own role models and imitate father’s behavior and vocabulary [59,60,61]. Moreover, McBride-Chang et al. followed 22 Chinese children from the beginning of kindergarten to Grade 1, and also found that the mediation of maternal guidance for children only explained children’s Chinese character reading, but not Chinese character writing [62].

### 4.4. The Pedagogical and Therapy Implication

The pedagogical implication of this study may be that it reveals the importance of specific cognitive abilities to the growth of Chinese written vocabulary size. The teachers should pay more attention to visual processing and meaning comprehension when formulating children’s education strategies. Meanwhile, due to boys and girls tending to use different cognitive abilities when learning Chinese characters, in the future, personalized and targeted teaching plans should be developed according to different genders. The present study also has a therapy implication for clinicians. Children with a delay in written vocabulary size generally have multiple cognitive skills deficits, alongside emotional and behavioral problems. In order to improve children’s difficulties in recognizing Chinese characters, clinicians should first focus on children’s cognitive deficits, rather than emotional and behavioral problems.

### 4.5. Limitations

First, the design of our study was a cross-sectional study which was not able to judge the causal relationship between the related factors and Chinese written vocabulary size. In the future, prospective studies are needed to confirm the results of our study. Second, the cognitive abilities of this study were assessed by parents filling out a questionnaire instead of a standardized neuropsychological experimental assessment, which may affect the accuracy of the results. However, the DCCC has good reliability and validity [27]. The cognitive abilities can be evaluated quickly and effectively while saving manpower and material resources.

## 5. Conclusions

In summary, when considering cognitive, emotional, and behavioral factors at the same time, there is a significant positive correlation between Chinese written vocabulary size and cognitive factors but not emotional and behavioral problems in primary school students, mainly including visual word recognition and meaning comprehension. Moreover, the related influencing factors of Chinese written vocabulary size were different between boys and girls. Boys are more dependent on meaning comprehension and spelling, while girls are more dependent on auditory and visual word recognition.

## Figures and Tables

**Table 1 ijerph-18-07797-t001:** Demographic information of the subjects (N = 1162).

	Total	Boys	Girls	*p*
	n (%)	n (%)	n (%)	
Sample size	1162 (100.0)	606 (52.2)	556 (47.8)	
Grade				
2	309 (26.6)	153 (25.2)	156 (28.1)	0.328
3	285 (24.5)	141 (23.3)	144 (25.9)	
4	303 (26.1)	166 (27.4)	137 (24.6)	
5	265 (22.8)	146 (24.1)	119 (21.4)	
Only one child				
Yes	519 (44.7)	278 (45.9)	241 (43.3)	0.386
No	643 (55.3)	328 (54.1)	315 (56.7)	
Mode of delivery				
Vaginal birth	639 (55.0)	335 (55.3)	304 (54.7)	0.836
Cesarean birth	523 (45.0)	271 (44.7)	252 (45.3)	
Father’s education level				
Junior middle school or below	253 (21.8)	138 (22.8)	115 20.7)	0.671
Senior middle school	320 (27.5)	161 (26.6)	159 (28.6)	
Junior college	264 (22.7)	142 (23.4)	122 (21.9)	
Bachelor degree or above	325 (28.0)	165 (27.2)	160 (28.8)	
Mother’s education level				
Junior middle school or below	298 (25.6)	153 (25.2)	145 (26.1)	0.982
Senior middle school	315 (27.1)	167 (27.6)	148 (26.6)	
Junior college	282 (24.3)	147 (24.3)	135 (24.3)	
Bachelor degree or above 6.5	267 (23.0)	139 (22.9)	128 (23.0)	
Family income (RMB/month/person)				
Less than 3000	141 (12.1)	61 (10.1)	80 (14.4)	0.074
3001~5000	279 (24.0)	145 (23.9)	134 (24.1)	
5001~10,000	334 (28.7)	192 (31.7)	142 (25.5)	
10,001~15,000	224 (19.3)	115 (19.0)	109 (19.6)	
More than 15,001	184 (15.8)	93 (15.3)	91 (16.4)	

**Table 2 ijerph-18-07797-t002:** Univariate analysis for related factors of Chinese written vocabulary size (N = 1162).

Variables	Chinese Written Vocabulary Size			A Delay in Written Vocabulary Size			
	$\bar{x}\;\pm \;s$	*t/F* */r*	*p*	Without $n\;(\%)/\;\bar{x}\;\pm \;s$	With $n\;(\%)/\;\bar{x}\;\pm \;s$	*t/χ* ^2^	*p*
**Demographic information**							
Sex		0.10	0.918			15.88	<0.001
Boys	2161.19 ± 773.77			544 (89.8)	62 (10.2)		
Girls				533 (95.9)	23 (4.1)		
Grade		1698.73	<0.001			0.51	0.917
2	1188.91 ± 285.35			289 (93.5)	20 (6.5)		
3	1940.60 ± 392.09			264 (92.6)	21 (7.4)		
4	2653.59 ± 333.94			279 (92.1)	24 (7.9)		
5	2959.44 ± 289.80			245 (92.5)	20 (7.5)		
Only one child		1.44	0.152			1.30	0.254
Yes	2194.64 ± 749.14			476 (91.7)	43 (8.3)		
No	2130.20 ± 770.34			601 (93.5)	42 (6.5)		
Mode of delivery		−1.14	0.255			3.92	0.048
Vaginal birth	2135.98 ± 763.88			601 (94.1)	38 (5.9)		
Cesarean birth	2187.08 ± 757.91			476 (91.0)	47 (9.0)		
Father’s education level		1.03	0.376			2.00	0.572
Junior middle school or below	2140.32 ± 795.18			231 (91.3)	22 (8.7)		
Senior middle school	2215.64 ± 736.60			300 (93.8)	20 (6.3)		
Junior college	2164.28 ± 756.39			242 (91.7)	22 (8.3)		
Bachelor degree or above	2113.40 ± 762.12			304 (93.5)	21 (6.5)		
Mother’s education level		0.86	0.462			2.69	0.441
Junior middle school or below	2159.39 ± 776.68			272 (91.3)	26 (8.7)		
Senior middle school	2197.62 ± 756.29			298 (94.6)	17 (5.4)		
Junior college	2172.46 ± 748.36			260 (92.2)	22 (7.8)		
Bachelor degree or above	2098.69 ± 763.98			247 (92.5)	20 (7.5)		
Family income (RMB/month/person)		2.59	0.035			4.63	0.327
Less than 3000	2057.40 ± 777.83			127 (90.1)	14 (9.9)		
3001~5000	2215.99 ± 767.07			263 (94.3)	16 (5.7)		
5001~10,000	2172.76 ± 770.89			313 (93.7)	21 (6.3)		
10,001~15,000	2227.21 ± 730.47			208 (92.9)	16 (7.1)		
More than 15,001	2042.28 ± 745.67			166 (90.2)	18 (9.8)		
**SDQ**							
Emotional symptom		−0.05 ^a^	0.078	1.91 ± 1.77	2.16 ± 1.75	−1.27	0.206
Conduct problem		−0.08 ^a^	0.010	1.74 ± 1.43	2.01 ± 1.31	−1.72	0.087
Hyperactivity/inattention		−0.12 ^a^	<0.001	3.96 ± 2.35	5.18 ± 2.34	−4.61	<0.001
Peer relationship problem		−0.002 ^a^	0.943	2.12 ± 1.56	2.66 ± 1.82	−2.64	0.010
Prosocial behavior		0.01 ^a^	0.631	7.36 ± 2.12	6.88 ± 2.50	1.73	0.087
**DCCC**							
The deficit of visual word recognition		−0.19 ^b^	<0.001	49.01 ± 9.54	55.35 ± 10.44	−5.86	<0.001
The deficit of auditory word recognition		−0.14 ^b^	<0.001	48.90 ± 9.53	55.00 ± 12.14	−4.53	<0.001
The deficit of meaning comprehension		−0.15 ^b^	<0.001	49.03 ± 9.63	55.52 ± 11.52	−5.06	<0.001
The deficit of spelling		−0.08 ^b^	0.010	49.12 ± 9.74	56.36 ± 11.94	−5.45	<0.001
The deficit of oral language		−0.12 ^b^	<0.001	49.13 ± 9.48	53.99 ± 11.92	−3.67	<0.001
The deficit of written expression		−0.08 ^b^	0.010	49.12 ± 9.74	56.36 ± 11.94	−5.45	<0.001
The deficit of attention		−0.14 ^b^	<0.001	49.05 ± 9.58	54.42 ± 11.11	−4.33	<0.001

Abbreviations: SDQ, Strengths and Difficulties Questionnaire; DCCC, The dyslexia checklist for Chinese children. Two-sample *t*-test, chi-square test, one-way ANOVA, and Pearson correlation analysis were used for univariate analysis. ^a^ The correlation coefficient between the subscale scores of SDQ and Chinese written vocabulary size. ^b^ The correlation coefficient between seven subscale T-scores of DCCC and Chinese written vocabulary size.

**Table 3 ijerph-18-07797-t003:** Multiple stepwise linear and logistic regression analysis for related factors of Chinese written vocabulary size (N = 1162).

Variables	Chinese Written Vocabulary Size, *β* (95% CI)			With a Delay in Written Vocabulary Size, *OR* (95% CI)		
	Model 1	Model 2	Model 3	Model 1	Model 2	Model 3
**Demographic information**						
Sex						
Boys		Ref			Ref	
Girls	60.73 ** (21.16, 100.30)	37.13 (−2.58, 76.84)	37.97 (−0.85, 76.78)	0.36 *** (0.22, 0.60)	0.43 ** (0.26, 0.72)	0.43 ** (0.26, 0.72)
Grade	617.22 *** (599.26, 635.17)	616.36 *** (598.64, 634.09)	612.64 *** (595.26, 630.03)			
2				Ref		
3				1.13 (0.59, 2.16)	1.10 (0.57, 2.13)	1.15 (0.59, 2.22)
4				1.14 (0.61, 2.15)	1.16 (0.61, 2.20)	1.27 (0.67, 2.43)
5				1.11 (0.57, 2.14)	1.08 (0.55, 2.11)	1.16 (0.59, 2.32)
Only one child						
Yes		Ref			Ref	
No	8.53 (−34.32, 51.37)	14.18 (−28.15, 56.52)	16.28 (−25.15, 57.70)	0.76 (0.47, 1.24)	0.73 (0.45, 1.19)	0.73 (0.45, 1.19)
Mode of delivery						
Vaginal birth		Ref			Ref	
Cesarean birth	−16.71 (−57.21, 23.79)	−13.63 (−53.50, 26.23)	−6.60 (−45.61, 32.41)	1.62 * (1.02, 2.58)	1.57 (0.98, 2.51)	1.51 (0.94, 2.43)
Father’s education level	48.40 *** (22.31, 74.49)	44.61 ** (18.92, 70.31)	36.00 ** (10.76, 61.25)			
Junior middle school or below				Ref		
Senior middle school				0.84 (0.40, 1.79)	0.84 (0.40, 1.78)	0.90 (0.42, 1.95)
Junior college				1.06 (0.45, 2.49)	1.09 (0.46, 2.58)	1.23 (0.51, 2.97)
Bachelor degree or above				0.74 (0.29, 1.88)	0.81 (0.32, 2.06)	0.90 (0.35, 2.35)
Mother’s education level	14.43 (−12.91, 41.76)	11.51 (−15.40, 38.43)	6.85 (−19.63, 33.33)			
Junior middle school or below				Ref		
Senior middle school				0.56 (0.26, 1.21)	0.65 (0.30, 1.40)	0.64 (0.29, 1.39)
Junior college				0.75 (0.32, 1.74)	0.84 (0.36, 1.94)	0.79 (0.34, 1.85)
Bachelor degree or above				0.76 (0.29, 1.99)	0.80 (0.30, 2.09)	0.81 (0.31, 2.13)
Family income (RMB/month/person)	−6.01 (−23.21, 11.19)	−8.52 (−25.47, 8.42)	−9.23 (−25.81, 7.35)			
Less than 3000				Ref		
3001~5000				0.51 (0.23, 1.09)	0.50 (0.23, 1.09)	0.47 (0.21, 1.04)
5001~10,000				0.58 (0.27, 1.26)	0.56 (0.26, 1.23)	0.52 (0.23, 1.16)
10,001~15,000				0.71 (0.31, 1.61)	0.77 (0.34, 1.75)	0.69 (0.30, 1.60)
More than 15,001				0.98 (0.43, 2.25)	1.03 (0.45, 2.40)	1.00 (0.42, 2.34)
**SDQ**						
Hyperactivity/inattention		−21.93 *** (−30.56, −13.30)	−2.79 (−12.63, 7.04)		1.18 ** (1.07, 1.30)	1.09 (0.98, 1.22)
Peer relationship problem		−17.41 ** (−30.15, −4.66)	−10.58 (−23.18, 2.01)		1.15 * (1.00, 1.32)	1.11 (0.97, 1.28)
**DCCC**						
The deficit of visual word recognition			−3.32 *** (−5.98, −0.66)			1.04 *** (1.02, 1.07)
The deficit of meaning comprehension			−6.52 * (−9.39, −3.64)			
**Adjusted R^2^ or Nagelkerke R^2^**	0.797	0.804	0.813	0.067	0.104	0.130

** p* < 0.05; ** *p* < 0.01; *** *p* < 0.001; abbreviations: CI, confidence interval; OR, odds ratio; Ref, reference group; SDQ, Strengths and Difficulties Questionnaire; DCCC, the dyslexia checklist for Chinese children. The demographic variables were forced into the regression models (Model 1), then the emotional and behavioral variables (five subscale scores of SDQ) were entered in a stepwise manner (Model 2), and then the cognitive variables (seven subscale T-scores of DCCC) were entered in a stepwise manner (Model 3), only retaining variables that are statistically significant in the model.

**Table 4 ijerph-18-07797-t004:** Subgroup analysis for related factors of Chinese written vocabulary size in different gender (N = 1162).

Variables	Chinese Written Vocabulary Size, *β* (95% CI)			With a Delay in Written Vocabulary Size, *OR* (95% CI)		
	Model 1	Model 2	Model 3	Model 1	Model 2	Model 3
**Boys (n = 606)**						
**Demographic information**						
Grade	623.93 *** (597.70, 650.16)	622.30 *** (596.44, 648.16)	618.36 *** (593.03, 643.68)			
2				Ref		
3				1.27 (0.60, 2.68)	1.26 (0.59, 2.69)	1.31 (0.61, 2.81)
4				0.98 (0.47, 2.07)	1.03 (0.49, 2.19)	0.98 (0.46, 2.10)
5				0.83 (0.37, 1.85)	0.84 (0.38, 1.88)	0.79 (0.35, 1.77)
Only one child						
Yes		Ref			Ref	
No	33.14 (−28.00, 94.27)	33.49 (−26.75, 93.72)	31.18 (−27.73, 90.10)	0.78 (0.44, 1.38)	0.78 (0.44, 1.38)	0.77 (0.43, 1.36)
Mode of delivery						
Vaginal birth		Ref			Ref	
Cesarean birth	−9.99 (−68.72, 48.74)	−8.31 (−66.18, 49.57)	−4.64 (−61.25, 51.97)	1.57 (0.91, 2.71)	1.53 (0.88, 2.65)	1.51 (0.87, 2.62)
Father’s education level	63.71 ** (26.09, 101.34)	57.86 ** (20.69, 95.03)	44.90 * (8.24, 81.56)			
Junior middle school or below				Ref		
Senior middle school				0.70 (0.30, 1.68)	0.69 (0.29, 1.65)	0.71 (0.29, 1.74)
Junior college				0.74 (0.27, 2.00)	0.73 (0.27, 1.99)	0.78 (0.28, 2.18)
Bachelor degree or above				0.49 (0.16, 1.49)	0.52 (0.17, 1.56)	0.56 (0.18, 1.71)
Mother’s education level	23.71 (−15.85, 63.27)	24.19 (−14.79, 63.17)	15.59 (−22.67, 53.84)			
Junior middle school or below				Ref		
Senior middle school				0.67 (0.28, 1.63)	0.79 (0.32, 1.91)	0.91 (0.37, 2.24)
Junior college				0.85 (0.31, 2.32)	0.93 (0.34, 2.52)	1.06 (0.39, 2.92)
Bachelor degree or above				0.91 (0.29, 2.85)	0.94 (0.30, 2.92)	1.18 (0.37, 3.70)
Family income (RMB/month/person)	−15.85 (−41.90, 10.21)	−17.35 (−43.03, 8.33)	−16.99 (−42.11, 8.12)			
Less than 3000				Ref		
3001~5000				0.69 (0.27.1.79)	0.69 (0.27, 1.81)	0.65 (0.25, 1.73)
5001~10,000				0.88 (0.34.2.26)	0.86 (0.33, 2.22)	0.85 (0.32, 2.26)
10,001~15,000				0.85 (0.30.2.36)	0.88 (0.32, 2.47)	0.84 (0.29, 2.42)
More than 15,001				1.06 (0.36.3.09)	1.11 (0.38, 3.26)	1.08 (0.36, 3.23)
**SDQ**						
Hyperactivity/inattention		−26.18 *** (−37.99, −14.36)	−7.01 (−20.56, 6.54)		1.17 ** (1.04, 1.31)	1.05 (0.91, 1.20)
**DCCC**						
The deficit of meaning comprehension			−8.96 *** (−12.27, −5.65)			
The deficit of spelling						1.04 ** (1.01, 1.07)
**Adjusted R^2^ or Nagelkerke R^2^**	0.784	0.791	0.800	0.029	0.052	0.080
**Girls (n = 556)**						
**Demographic information**						
Grade	611.06 *** (586.62, 635.50)	610.06 *** (585.87, 634.26)	609.51 *** (585.80, 633.22)			
2				Ref		
3				0.73 (0.18, 2.86)	0.56 (0.14, 2.33)	0.59 (0.14, 2.58)
4				1.73 (0.52, 5.79)	1.46 (0.42, 5.10)	1.93 (0.55, 6.82)
5				2.15 (0.64, 7.26)	2.16 (0.60, 7.76)	2.70 (0.70, 10.44)
Only one child						
Yes		Ref			Ref	
No	−24.48 (−84.56, 35.59)	−16.97 (−76.44, 42.50)	−2.21 (−60.79, 56.37)	0.81 (0.31, 2.10)	0.72 (0.27, 1.91)	0.69 (0.25, 1.89)
Mode of delivery						
Vaginal birth		Ref			Ref	
Cesarean birth	−27.03 (−82.70, 28.64)	−22.94 (−77.82, 31.94)	−21.94 (−75.72, 31.83)	2.03 (0.81, 5.08)	1.93 (0.74, 5.04)	1.89 (0.71, 5.02)
Father’s education level	29.67 (−6.42, 65.77)	27.67 (−7.91, 63.25)	28.20 (−6.67, 63.06)			
Junior middle school or below				Ref		
Senior middle school				1.74 (0.37, 8.17)	1.78 (0.36, 8.88)	1.81 (0.35, 9.39)
Junior college				3.69 (0.61, 22.16)	4.29 (0.69, 26.84)	4.61 (0.71, 29.97)
Bachelor degree or above				2.56 (0.37, 17.51)	3.41 (0.49, 23.84)	3.26 (0.44, 24.45)
Mother’s education level	5.19 (−32.41, 42.78)	−0.64 (−37.77, 36.49)	−4.70 (−41.12, 31.72)			
Junior middle school or below				Ref		
Senior middle school				0.27 (0.05, 1.36)	0.35 (0.06, 1.89)	0.35 (0.06, 2.06)
Junior college				0.51 (0.11, 2.48)	0.59 (0.12, 2.93)	0.58 (0.11, 3.04)
Bachelor degree or above				0.52 (0.09, 3.12)	0.68 (0.11, 4.16)	0.71 (0.11, 4.71)
Family income (RMB/month/person)	3.30 (−19.25, 25.85)	0.71 (−21.53, 22.96)	−3.01 (−24.86, 18.83)			
Less than 3000				Ref		
3001~5000				0.22 * (0.05, 0.97)	0.20 * (0.04, 0.93)	0.16 * (0.03, 0.79)
5001~10,000				0.12 * (0.02, 0.71)	0.11 * (0.02, 0.68)	0.07 ** (0.01, 0.51)
10,001~15,000				0.48 (0.12, 2.02)	0.59 (0.13, 2.67)	0.50 (0.11, 2.38)
More than 15,001				0.74 (0.19, 2.94)	0.88 (0.21, 3.73)	0.78 (0.17, 3.53)
**SDQ**						
Hyperactivity/inattention		−18.75 ** (−31.20, −6.30)	−3.26 (−16.97, 10.44)		1.32 * (1.07, 1.64)	1.18 (0.93, 1.49)
Peer relationship problem		−20.79 * (−38.56, −3.02)	−11.97 (−29.74, 5.80)		1.32 * (1.00, 1.74)	1.24 (0.92, 1.66)
**DCCC**						
The deficit of visual word recognition						1.07 ** (1.02, 1.12)
The deficit of auditory word recognition			−8.86 *** (−12.44, −5.29)			
**Adjusted R^2^ or Nagelkerke R^2^**	0.813	0.819	0.826	0.121	0.205	0.253

* *p* < 0.05; ** *p* < 0.01; *** *p* < 0.001; abbreviations: CI, confidence interval; OR, odds ratio; Ref, reference group; SDQ, Strengths and Difficulties Questionnaire; DCCC, the dyslexia checklist for Chinese children. The demographic variables were forced into the regression models (Model 1), then the emotional and behavioral variables (five subscale scores of SDQ) were entered in a stepwise manner (Model 2), and then the cognitive variables (seven subscale T-scores of DCCC) were entered in a stepwise manner (Model 3), only retaining variables that are statistically significant in the model.

## Data Availability

We will not share our raw data in the manuscript, because the raw data of this article is being applied to another unpublished article and the project group has not published the data in whole elsewhere.

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
