# Peer review of "The Correlation between Chinese Written Vocabulary Size and Cognitive, Emotional and Behavioral Factors in Primary School Students"

_ijerph, 2021, doi:10.3390/ijerph18157797_

Round 1
Reviewer 1 Report
The topic of the manuscript addresses the examination of the correlation between Chinese vocabulary size and cognitive, emotional and behavioural factors in primary school students. The research is particularly interesting because of the growing interest in the development of reading comprehension skills at early ages as well as the factors that are associated with the growth of children’s vocabulary size.
Despite the growing number of studies on this issue, few researchers so far have focused their attention on the relationship between vocabulary size and children’s characteristics, including cognitive, emotional and behavioural factors. Thus, the study is potentially very valuable.
Nevertheless, there are several points that I believe need some improvement. The theoretical background is not sufficiently developed. In this respect, I missed a more comprehensive description of what has been done so far in this area, and in which way the study contributes to filling the existing gap. I would also appreciate the authors clarify what exactly they mean by cognitive, emotional and behavioural factors. It could be a good idea to include the research questions as well.
In respect of the methodology, I would propose to amplify the methodological section, providing a more consistent description of the procedure of the study.
The discussion section could be better organized. Please, refer to your research questions and the results you have managed to achieve. Discuss in which way the results of this study relates to the results of other similar research
I would also recommend specifying in a better way the limitations and the pedagogical implications of the study.
Finally, please, take into consideration that the whole manuscripts require additional proofreading.
Reviewer 2 Report
Please, check the manuscript (att.).
- define the vocabulary size -do you mean written VC?
- add information regarding the sample (bi-mono-plurilingual; Asian people only?)
- Please, comment the difference between the father's and mother's level of education and their impact on the vocabulary.
- add implications for therapy
The papr is very interesting. Thank you!
